# Factors Associated with Procedural Thromboembolisms after Mechanical Thrombectomy for Acute Ischemic Stroke

**DOI:** 10.3390/medicina56070353

**Published:** 2020-07-16

**Authors:** Taek Min Nam, Ji Hwan Jang, Young Zoon Kim, Kyu Hong Kim, Seung Hwan Kim

**Affiliations:** Department of Neurosurgery, Samsung Changwon Hospital, Sungkyunkwan University School of Medicine, Changwon 51353, Korea; taekmin82@gmail.com (T.M.N.); gebassist@naver.com (J.H.J.); yzkim@skku.edu (Y.Z.K.); Unikkh@unitel.co.kr (K.H.K.)

**Keywords:** acute ischemic stroke, mechanical thrombectomy, procedural thromboembolisms

## Abstract

*Background and objective:* Procedural thromboembolisms after mechanical thrombectomy (MT) for acute ischemic stroke has rarely been studied. We retrospectively evaluated factors associated with procedural thromboembolisms after MT using diffusion-weight imaging (DWI) within 2 days of MT. *Materials and Methods:* From January 2018 to March 2020, 78 patients with acute ischemic stroke who underwent MT were evaluated using DWI. Procedural thromboembolisms were defined as new cerebral infarctions in other territories from the occluded artery on DWI after MT. *Results:* Procedural thromboembolisms were observed on DWI in 16 patients (20.5%). Procedural thromboembolisms were associated with old age (73.8 ± 8.18 vs. 66.8 ± 11.2 years, *p* = 0.021), intravenous (IV) thrombolysis (12 out of 16 (75.0%) vs. 25 out of 62 (40.3%), *p* = 0.023), heparinization (4 out of 16 (25.0%) vs. 37 out of 62 (59.7%), *p* = 0.023), and longer procedural time (90.9 ± 35.6 vs. 64.4 ± 33.0 min, *p* = 0.006). Multivariable logistic regression analysis revealed that procedural thromboembolisms were independently associated with procedural time (adjusted odds ratio (OR); 1.020, 95% confidence interval (CI); 1.002–1.039, *p* = 0.030) and IV thrombolysis (adjusted OR; 4.697, 95% CI; 1.223–18.042, *p* = 0.024). The cutoff value of procedural time for predicting procedural thromboembolisms was ≥71 min (area under the curve; 0.711, 95% CI; 0.570–0.851, *p* = 0.010). *Conclusions:* Procedural thromboembolisms after MT for acute ischemic stroke are significantly associated with longer procedural time and IV thrombolysis. This study suggests that patients with IV thrombolysis and longer procedural time (≥71 min) are at a higher risk of procedural thromboembolisms after MT for acute ischemic stroke.

## 1. Introduction

Mechanical thrombectomy (MT) of a major vessel occlusion of the anterior circulation has become a standard treatment for acute ischemic stroke. Several randomized controlled trials (RCTs) have shown the benefits of this procedure [1,2,3,4,5]. Due to recent developments in interventional devices and techniques, endovascular procedures have been considered to be safer and faster treatment options for the treatment of cerebrovascular disease in patients with large vessel occlusion. However, procedural thromboembolisms are still one of the major complications of endovascular procedures.

Several studies have evaluated the risk factors of procedural thromboembolisms using diffusion-weighted imaging (DWI) after endovascular procedures. Procedural thromboembolisms after endovascular procedures are associated with the type and dose of antiplatelet drug, resistance to the antiplatelet drug, the procedural time, the number of DWI-positive lesions, and the type of device used [6,7,8,9]. However, these previous studies on procedural thromboembolisms have focused on coil embolization for cerebral aneurysms and carotid artery stenting for carotid artery stenosis.

As recanalization is the major goal of MT for acute ischemic stroke, procedural thromboembolisms have not been considered as important as recanalization, and procedural thromboembolisms after MT for acute ischemic stroke has rarely been studied. In this study, we aimed to evaluate factors affecting procedural thromboembolisms after MT for acute ischemic stroke using DWI.

## 2. Materials and Methods

### 2.1. Patient Selection Criteria

We reviewed the data of patients with acute ischemic stroke who underwent MT at our hospital from January 2018 to March 2020. A total of 139 patients who met the inclusion criteria were selected for this study. The inclusion criteria were as follows: (1) acute ischemic stroke with symptoms; (2) large artery occlusions confirmed by magnetic resonance (MR) angiography; (3) ≤24 h from the last normal time (LNT) to treatment; (4) at least one-half mismatch between cerebral blood flow and cerebral blood volume map with MR perfusion imaging; (5) treatment with MT.

Of the 139 patients, 61 patients were excluded owing to the lack of data on DWI within 2 days of MT. Therefore, only 78 consecutive patients were included. The case accrual process is summarized in Figure 1.

### 2.2. Data Collection and Patient Characteristics

The data on the baseline characteristics of the patients, treatment details, and clinical and radiological outcomes were obtained from medical records. The medical records and imaging data were reviewed after approval from the institutional review board (SCMC 2020-06-006, approved date 17 June 2020). The baseline characteristics of the patients included age; sex; past medical history, including atrial fibrillation, diabetes mellitus, hypertension, and previous cerebral infarction; smoking status; stroke etiology; occlusion site; tandem occlusion; National Institutes of Health Stroke Scale (NIHSS) score on admission; as well as the time from the LNT to puncture. Treatment details included the use of intravenous (IV) thrombolysis, antiplatelet preparation, heparinization before initiating MT, use of balloon-guided catheter, use of intermediate catheter, total number of MT attempts, procedural time, and whether rescue treatments, such as angioplasty or stenting, were performed. The clinical and radiological outcomes included a 3-month modified Rankin Scale (mRS) score, thrombolysis in cerebral infarction (TICI) grade, incidence of procedural thromboembolisms, and incidence of postprocedural hemorrhages. Among the clinical and radiological outcomes, the incidence of procedural thromboembolism is the primary outcome in this study.

Stroke etiology was classified according to the trial of ORG 10,172 in acute stroke treatment (TOAST) criteria. Each patient’s NIHSS score was assessed on admission. Patients admitted within 4.5 h of acute ischemic stroke symptom onset were considered suitable candidates for IV tissue-plasminogen activator (t-PA) infusion. Patients who were administered aspirin or clopidogrel were included in the antiplatelet preparation group, while those who were administered neither aspirin nor clopidogrel were included in the no antiplatelet preparation group. When IV thrombolysis was not performed, bolus doses of heparin (2000–3000 IU) were administered before initiating MT, followed by a continuous infusion of heparin (500 IU/h). When IV thrombolysis was performed, heparin was not administered due to the concern of intracerebral hemorrhage. The procedural time was defined as the total time from a femoral puncture to recanalization. A good clinical outcome was defined as a 3-month mRS score of ≤2. Successful recanalization was defined as a TICI grade of 2b or 3. Procedural thromboembolisms were defined as new cerebral infarctions in other territories from the occluded artery on DWI after MT. Postprocedural hemorrhages included subarachnoid hemorrhage and intracerebral hemorrhage (except for hemorrhagic transformation), based on the European cooperative acute stroke study (ECASS) classification [10].

### 2.3. Procedures

The procedures were performed under conscious sedation via femoral access. The primary MT modality was decided based on the surgeon’s discretion and performed using catheter aspiration or a stent retriever. The use of a balloon-guided catheter (Flowgate, Stryker Neurovascular, Fremont, CA, USA) and intermediate catheter (Catalyst 6, Stryker Neurovascular, Mountain View, CA, USA) were decided based on the condition of the vessels. When the angiography after MT showed residual stenosis in atherosclerotic occlusions, rescue treatments such as balloon angioplasty or stenting were considered.

### 2.4. Statistical Analysis

The baseline characteristics of the patients, treatment details, and clinical and radiological outcomes were compared between the procedural thromboembolism group and the no procedural thromboembolism group. The chi-squared test or Fisher’s exact test was used to analyze categorical variables. Student’s *t*-test or the Mann–Whitney U test was used for continuous variables. Multivariable analysis, including logistic regression, was used to evaluate factors affecting procedural thromboembolisms after MT for acute ischemic stroke. Variables with *p* < 0.05 in the univariate analysis were selected for the logistic regression model. The cutoff value of the procedural time for predicting procedural thromboembolisms was obtained using the receiver—operating characteristic (ROC) curve. A *p*-value < 0.05 was considered statistically significant. All analyses were performed with SPSS version 22 (SPSS, Chicago, IL, USA).

## 3. Results

The baseline characteristics and treatment details of the included and excluded patients are summarized in Table 1. None of the baseline characteristics and treatment details, except for procedure time, differed significantly between the groups. Procedural time was significantly longer for the included patients than for the excluded patients (69.8 ± 35.0 vs. 58.2 ± 31.2 min, *p* = 0.043).

The mean age of the patients was 66.5 (range 33–87) years. Of the 78 patients, 51 were males. The stroke etiologies were as follows: large-artery atherosclerosis (n = 23), cardioembolism (n = 30), other determined etiology (n = 1), and undetermined etiology (n = 24). The sites of occlusion were as follows: anterior circulations (n = 72), including cervical to cavernous internal carotid artery (ICA) occlusion (n = 7), cervical ICA occlusion with middle cerebral artery occlusion (n = 8), ICA bifurcation occlusion (n = 9), M1 occlusion (n = 40), and M2 occlusion (n = 8), as well as posterior circulations (n = 6), including basilar artery occlusion (n = 4), and vertebral artery occlusion (n = 2).

Among the 78 patients who underwent MT for acute ischemic stroke, procedural thromboembolisms were observed in 16 (20.5%) patients on DWI within 2 days of MT; all of them were small and focal cerebral infarctions. The new cerebral infarctions on DWI after MT were located in the cortex of the contralateral middle cerebral artery (MCA) territory (n = 9) and cerebellum (n = 7). The total number of new cerebral infarctions on DWI was as follows: 1 (n = 8), 2 (n = 4), and 3 (n = 4). A case of MT for acute ischemic stroke and procedural thromboembolisms is shown in Figure 2.

The baseline characteristics of the patients, treatment details, and clinical and radiological outcomes of the procedural thromboembolism and no procedural thromboembolism groups are summarized in Table 2. None of the baseline characteristics, except for age, differed significantly between the groups. Patients in the procedural thromboembolism group were significantly older than those in the no procedural thromboembolism group (73.8 ± 8.18 vs. 66.8 ± 11.2 years, *p* = 0.021).

Procedural thromboembolisms had a significant relationship with the use of IV thrombolysis, heparinization before initiating MT, and procedural time. IV thrombolysis was more frequently used in the procedural thromboembolism group than in the no procedural thromboembolism group (12 out of 16 (75%) vs. 25 out of 62 (40.3%), *p* = 0.023). Heparinization before initiating MT was less frequently used in the procedural thromboembolism group than in the no procedural thromboembolism group (4 out of 16 (25.0%) vs. 37 out of 62 (59.7%), *p* = 0.023). The procedural time was longer in the procedural thromboembolism group than in the no procedural thromboembolism group (90.9 ± 35.6 vs. 64.4 ± 33.0 min, *p* = 0.006).

The mRS scores were as follows: mRS 0 (n = 15), mRS 1 (n = 16), mRS 2 (n = 16), mRS 3 (n = 10), mRS 4 (n = 7), mRS 5 (n = 6), and mRS 6 (n = 8). The TICI grades were as follows: TICI 0 (n = 4), TICI 1 (n = 2), TICI 2a (n = 9), TICI 2b (n = 8), and TICI 3 (n = 55). Good clinical outcomes and successful recanalization did not differ significantly between the groups.

Postprocedural hemorrhage was observed in 20 (25.6%) patients, including subarachnoid hemorrhage (n = 11) and intracerebral hemorrhage (n = 9). The incidence of postprocedural hemorrhage was higher in the procedural thromboembolism group than in the no procedural thromboembolism group (9 out of 16 (56.3%) vs. 11 out of 62 (17.7%), *p* = 0.003).

The multivariable logistic regression analysis for factors affecting procedural thromboembolisms after MT for acute ischemic stroke, including age, IV thrombolysis, heparinization, and procedural time, showed that procedural time (adjusted odds ratio (OR); 1.020, 95% confidence interval (CI); 1.002–1.039, *p* = 0.030) and IV thrombolysis (adjusted OR; 4.697, 95% CI; 1.223–18.042, *p* = 0.024) were independently associated with procedural thromboembolisms after MT for acute ischemic stroke (Table 3).

ROC curve analysis for procedural thromboembolisms during MT for acute ischemic stroke is shown in Table 4. The ROC curve analysis showed that the cutoff value of procedural time for predicting procedural thromboembolisms was ≥71 min (area under the curve (AUC); 0.711, 95% CI; 0.570–0.851, *p* = 0.010). In total, 33 out of 78 (42.3%) patients had a procedural time of ≥71 min. Among patients with procedural thromboembolisms, 12 out of 16 (75.0%) patients had a procedural time of ≥71 min. The ROC curve analysis also showed that procedural time and IV thrombolysis were most significantly associated with procedural thromboembolisms (AUC; 0.795, 95% CI; 0.677–0.914, *p* ≤ 0.001).

## 4. Discussion

MT has been considered the standard treatment option for acute ischemic stroke and has many theoretical advantages, including site specificity and high recanalization rates. Previous studies have focused on the clinical and radiological outcomes after MT, such as functional outcomes, recanalization rates, and hemorrhagic complications [1,2,3,4,5]. However, procedural thromboembolism after MT for acute ischemic stroke has not been studied. This study assessed the risk factors of procedural thromboembolism after MT for acute ischemic stroke.

In this study, a total of 16 patients (20.5%) had procedural thromboembolisms. This result is in line with that of previous reports, in which the rate of procedural thromboembolisms during coil embolization for cerebral aneurysms and carotid artery stenting for carotid artery stenosis was 10–60% [8,9,11].

Cerebral infarction on DWI after neurointerventional procedure may be caused by tiny thrombi, fragmented atherosclerotic plaques, air bubbles or hydrophilic coating materials from catheters and wires during catheter insertion or injection of contrast media or flushing saline [12,13]. Our multivariate analysis showed a significant relationship between procedural thromboembolisms and procedural time (adjusted OR; 1.020, 95% CI; 1.002–1.039, *p* = 0.030). Longer procedural time is associated with a higher chance of dislodging a thrombus and introducing air bubbles or hydrophilic coating materials during the procedure [7].

Previous studies have reported that a procedural time of >60 min or three retrieval attempts during MT can lead to a reduced rate of good clinical outcomes and increased risk of hemorrhage [14,15]. In our study, the cutoff value of the procedural time in predicting procedural thromboembolisms was ≥71 min, suggesting that a procedural time of >1 h is associated with a decreased rate of good clinical outcome and increased risk of hemorrhage and procedural thromboembolism.

The procedural time can be longer owing to the technical difficulties encountered during the procedure, such as advancing the guiding catheter into the carotid artery or subclavian artery [16]. Advancing a guiding catheter can be more difficult for patients with acute ischemic stroke than for those with other neurointerventional procedures, because most patients are elderly with substantial comorbidities and poor vascular health with widespread atheroma, calcification, tortuous vessels, and occlusive disease [17]. This study showed no significant difference in the total number of MT attempts between the two groups (3.38 ± 2.78 vs. 2.79 ± 2.24 times, *p* = 0.379); however, the procedural time was significantly different (90.9 ± 35.6 vs. 64.4 ± 33.0 min, *p* = 0.006) between the groups. This can be partially explained with the age difference between the two groups (73.8 ± 8.18 vs. 66.8 ± 11.2 years, *p* = 0.021), suggesting a possibility of technical difficulty in advancing the guiding catheter. Our results also suggest the importance of setting an optimal strategy and preparing devices for advancing the guiding catheter before the procedure to reduce the procedural time.

In our study, IV thrombolysis was independently associated with procedural thromboembolisms after MT for acute ischemic stroke (adjusted OR; 4.697, CI; 1.223–18.042, *p* = 0.024). IV thrombolysis with t-PA has been the therapy of choice for acute ischemic stroke within a 4.5-h time window for 20 years. Recent RTCs have proven that MT with IV thrombolysis is superior to IV thrombolysis alone in patients with acute ischemic stroke caused by large artery occlusion in the anterior circulation [4,18,19]. IV thrombolysis with t-PA before MT improves the clinical outcomes after MT by enhancing the fibrinolytic process, increasing the speed and likelihood of successful reperfusion with MT, reducing the required number of passes, and decreasing the frequency of microvascular thrombus [19].

Among these effects of IV thrombolysis with t-PA, decreasing the frequency of microvascular thrombi is thought to be due to the antithrombotic effects of t-PA. However, our results showed that IV thrombolysis was independently associated with procedural thromboembolisms after MT for acute ischemic stroke. The antithrombotic effect of t-PA during MT has not been studied, and there have been no reports on the relationship between IV thrombolysis and procedural thromboembolisms after MT for acute ischemic stroke.

In this study, the definition of procedural thromboembolisms was new cerebral infarctions in other territories from the occluded artery. Therefore, procedural thromboembolisms were different from the thrombus migration of fragmentation, even though there was a possibility that the thromboembolisms originated from the collateral flow from non-targeted vessels [20]. New cerebral infarctions in other territories from the occluded artery may originate from the atherosclerotic plaques scraped off the aortic arch or great vessel wall by catheter manipulation [21].

IV t-PA before MT is associated with enhancing successful recanalization owing to its fibrinolytic effect [18]. However, in a recent study, IV thrombolysis with t-PA before MT resulted in thrombus fragmentation and more MT attempts [22]. Similarly, IV thrombolysis with t-PA before MT may have an antithrombotic effect and prevent procedural thromboembolisms owing to its fibrinolytic effect. However, it can also promote procedural thromboembolisms by inducing embolization of atherosclerotic plaques from the aortic arch, as seen in our study.

In this study, all procedural thromboembolisms were found to be small and focal cerebral infarctions on DWI. A previous study has shown that a larger number of DWI-positive lesions (n ≥ 6) is significantly related to symptomatic ischemic complication [6]. In our results, the total number of DWI-positive lesions was ≤3 in all patients with procedural thromboembolisms, which can partially explain why procedural thromboembolism was not significantly related to clinical outcome. Another study has shown that asymptomatic cerebrovascular brain injury may be associated with cognitive impairment, stroke, and mortality [23]. Therefore, it is crucial to reduce the incidence of procedural thromboembolisms and identify the underlying causes of postprocedural cerebral infarction, including asymptomatic cerebral infarction, during neurointerventional procedures.

In this study, postprocedural hemorrhage was significantly associated with procedural thromboembolisms (56.3% vs. 17.7%, *p* = 0.003). Our result showed the increased risk of both thrombosis and bleeding in patients who underwent MT for acute ischemic stroke. The mechanism of this imbalance is not clear. A previous study reported that coagulation tests had little value in the evaluation of general patients with ischemic stroke [24]. However, further studies with circulating biomarkers, such as circulating inflammatory factors and coagulation factors, might help explain the duality of thrombosis and bleeding in patients who underwent MT for acute ischemic stroke. Further studies are needed to confirm these preliminary results.

Several limitations of this study should be noted. First, this study has a retrospective design with attendant selection bias. However, the selection bias may be low because all cases of MT in our hospital were included. Second, due to the very small sample size, profound statistical analysis could not be obtained, and risk factors for procedural thromboembolisms may be underestimated. Third, in our study, procedural thromboembolisms were defined as new cerebral infarctions in other territories from the occluded artery on DWI after MT. This definition did not include new cerebral infarctions in the entire brain area. Since distinguishing a new cerebral infarction in the same territory of the occluded artery from migration and fragmentation of the thrombus is not easy, we used our definition, which could represent less than real procedural thromboembolisms. Fourth, we did not include late thromboembolisms. This study focused only on the analysis of acute procedural thromboembolisms using DWI after MT. In our hospital, magnetic resonance imaging (MRI) including DWI and MR angiography was performed to evaluate postprocedural hemorrhage and reocclusion on the first to second day after MT. This study is a retrospective analysis using DWI from these results. This study may underestimate procedural thromboembolisms because it does not include the late thromboembolisms that occur later than 2 days. Further studies with larger sample sizes and long-term follow-up are needed to confirm these preliminary results.

## 5. Conclusions

Procedural thromboembolisms after MT for acute ischemic stroke are significantly associated with longer procedural time and IV thrombolysis. This study suggests that patients with IV thrombolysis and longer procedural time (≥71 min) are at a higher risk of procedural thromboembolisms after MT for acute ischemic stroke. Further studies with larger sample sizes and long-term follow-up are needed to confirm these preliminary results.

## Figures and Tables

**Figure 1 medicina-56-00353-f001:**
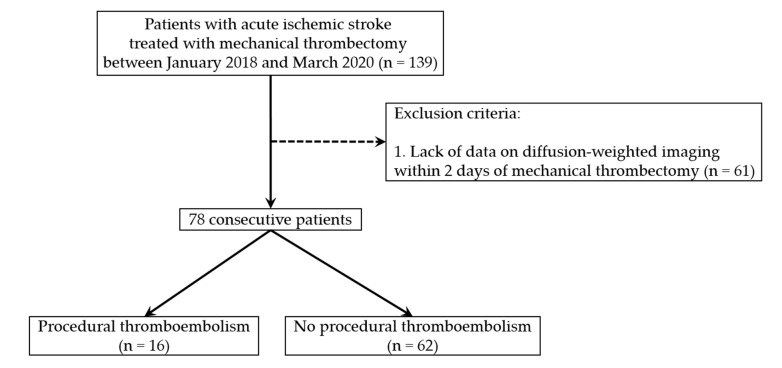
Flowchart of the case accrual process.

**Figure 2 medicina-56-00353-f002:**
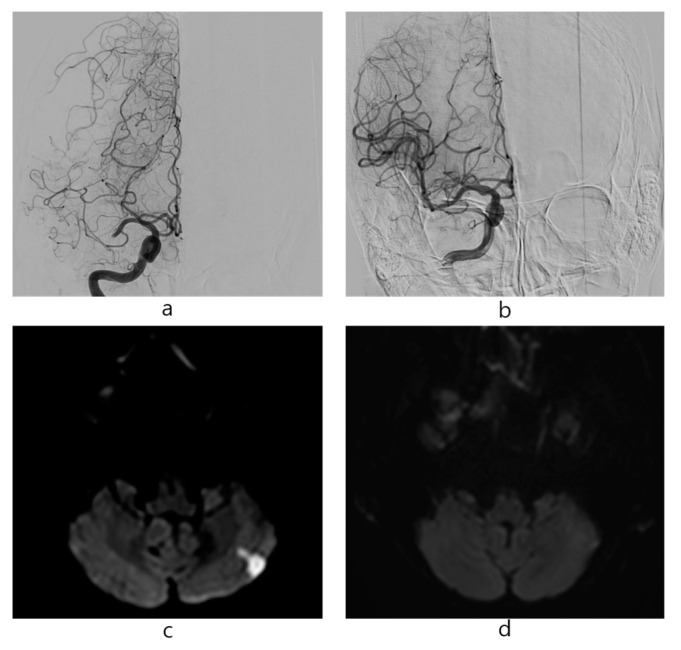
A case of mechanical thrombectomy for right M1 occlusion. An 81-year-old female patient presented with left hemiparesis. (**a**) Proximal M1 occlusion was observed. (**b**) Successful recanalization was achieved in M1 occlusion using a stent retriever. (**c**) Diffusion-weighted imaging 1 day after mechanical thrombectomy showed a new lesion in the left cerebellum, (**d**) which was not shown in diffusion-weighted imaging before mechanical thrombectomy.

**Table 1 medicina-56-00353-t001:** A comparison of baseline characteristics and treatment details between included and excluded patients.

		Included Patients (n = 78)	Excluded Patients (n = 61)	*p*-Value
Baseline characteristics	Age (years) (mean ± SD)	68.2 ± 10.9	67.1 ± 14.9	0.597
	Sex (male)	51 (65.4%)	41 (67.2%)	0.858
	Atrial fibrillation	28 (35.9%)	25 (41.0%)	0.599
	Diabetes Mellitus	16 (20.5%)	14 (23.0%)	0.836
	Hypertension	45 (57.7%)	34 (55.7%)	0.864
	Previous cerebral infarction	9 (11.5%)	10 (16.4%)	0.461
	Smoking	28 (35.9%)	22 (36.1%)	1.000
	Stroke etiology			0.602
	Cardioembolic	30	27	
	Non-cardioembolic	48	34	
	Occlusion site (anterior circulation)	73 (93.6%)	54 (88.5%)	0.366
	Tandem occlusion	10 (12.8%)	6 (9.8%)	0.790
	NIHSS score on admission (mean ± SD)	11.3 ± 5.88	12.6 ± 7.65	0.240
	The time from the LNT to the puncture (mean ± SD)	371.0 ± 246.2	343.7 ± 230.3	0.505
Treatment details	IV thrombolysis	37 (47.4%)	20 (32.8%)	0.086
	Antiplatelet preparation	17 (21.8%)	20 (32.8%)	0.177
	Heparinization	41 (52.6%)	41 (67.2%)	0.086
	Balloon-guided catheter	15 (19.2%)	20 (32.8%)	0.078
	Intermediate catheter	69 (88.5%)	46 (75.4%)	0.069
	Total number of MT attempts (mean ± SD)	2.91 ± 2.35	2.43 ± 1.64	0.174
	Procedural time (mean ± SD)	69.8 ± 35.0	58.2 ± 31.2	0.043
	Rescue treatment	9 (11.5%)	6 (9.8%)	0.790

NIHSS: National Institutes of Health Stroke Scale; LNT: last normal time; SD: standard deviation; MT: mechanical thrombectomy; IV: intravenous.

**Table 2 medicina-56-00353-t002:** A comparison of baseline characteristics, treatment details, and clinical and radiological outcomes between the procedural thromboembolism group and no procedural thromboembolism group after mechanical thrombectomy.

		Procedural Thromboembolism		*p*-Value
		Yes (n = 16)	No (n = 62)	
Baseline characteristics	Age (years) (mean ± SD) *	73.8 ± 8.18	66.8 ± 11.2	0.021
	Sex (male)	10 (62.5%)	41 (66.1%)	0.776
	Atrial fibrillation	8 (50.0%)	20 (32.3%)	0.245
	Diabetes Mellitus	4 (25.0%)	12 (19.4%)	0.729
	Hypertension	10 (62.5%)	35 (56.5%)	0.780
	Previous cerebral infarction	0	9 (14.5%)	0.191
	Smoking	3 (18.8%)	25 (40.3%)	0.147
	Stroke etiology			0.149
	Cardioembolic	9	21	
	Non-cardioembolic	7	41	
	Occlusion site (anterior circulation)	16 (100%)	57 (91.9%)	0.577
	Tandem occlusion	3 (18.8%)	7 (11.3%)	0.420
	NIHSS score on admission (mean ± SD)	11.5 ± 5.32	11.2 ± 6.05	0.861
	The time from the LNT to the puncture (mean ± SD)	305.0 ± 223.7	388.1 ± 250.6	0.231
Treatment details	IV thrombolysis *	12 (75.0%)	25 (40.3%)	0.023
	Antiplatelet preparation	2 (12.5%)	15 (24.2%)	0.499
	Heparinization *	4 (25.0%)	37 (59.7%)	0.023
	Balloon-guided catheter	1 (6.3%)	14 (22.6%)	0.175
	Intermediate catheter	16 (100%)	53 (85.5%)	0.191
	Total number of MT attempts (mean ± SD)	3.38 ± 2.78	2.79 ± 2.24	0.379
	Procedural time (mean ± SD) *	90.9 ± 35.6	64.4 ± 33.0	0.006
	Rescue treatment	2 (12.5%)	7 (11.3%)	1.000
Clinical and radiological outcomes	Good clinical outcome	11 (68.8%)	36 (58.1%)	0.570
	Successful recanalization	14 (87.5%)	49 (79.0%)	0.723
	Postprocedural hemorrhage	9 (56.3%)	11 (17.7%)	0.003

NIHSS: National Institutes of Health Stroke Scale; LNT: last normal time; SD: standard deviation. * Variables for logistic regression model.

**Table 3 medicina-56-00353-t003:** Multivariable logistic regression analysis of factors associated with procedural thromboembolisms during MT for acute ischemic stroke.

	Adjusted OR	Adjusted 95% CI	*p*-Value
Age	1.084	0.992–1.185	0.076
Procedural time	1.020	1.002–1.039	0.030
IV thrombolysis	4.697	1.223–18.042	0.024

MT: Mechanical thrombectomy; OR: odds ratio; CI: confidence interval; IV: intravenous.

**Table 4 medicina-56-00353-t004:** ROC curve analysis for procedural thromboembolisms during MT for acute ischemic stroke.

	AUC	95% CI	*p*-Value
Procedural time	0.711	0.570–0.851	0.010
IV thrombolysis	0.634	0.483–0.785	0.077
Procedural time and IV thrombolysis	0.795	0.677–0.914	<0.001

ROC: receiver operating characteristic; MT: mechanical thrombectomy; AUC: area under the curve; OR: odds ratio; CI: confidence interval.

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
