# Peer review of "Factors Associated with Procedural Thromboembolisms after Mechanical Thrombectomy for Acute Ischemic Stroke"

_medicina, 2020, doi:10.3390/medicina56070353_

Round 1
Reviewer 1 Report
In the article “Factors associated with procedural thromboembolisms after mechanical thrombectomy for acute ischemic stroke” the authors show in a retrospective study that procedural thromboembolisms after MT for acute ischemic stroke are significantly associated with longer procedural time and IV thrombolysis.
Although the study is interesting and well performed, the findings are quite predictable and expected.
Major points:
- The finding that IV thrombolysis is associated with procedural thromboembolism is interesting. To underline the findings the authors should measure plasma levels of circulating biomarkers like e.g. A2M, SAP and especially MMP-9. Is there a difference in the patient groups with and without IV thrombolysis? Does it translate to different biomarker levels between the tPA treated patients developping a embolism and the ones which do not develop procedural thromboembolisms. This correlation would significantly strengthen the manuscript.
- In addition, the authors state in line 228 that tPa could also have fibrinolytic effects thus effecting embolism. To rule this out the plasma levels of D-dimer in the corresponding patient cohorts could be determined.
Minor points:
Please add "This result is in line..." in line 175.
Reviewer 2 Report
In this study the authors investigated the occurrence of new ischemic lesions in patients undergoing MT for AIS. Differently from previous studies, exploring procedural thromboembolism in the same territory of the occluded artery from migration or fragmentation of the thrombus, they analyzed new ischemic lesions in other territories. The rate of procedural thromboembolism was 20.5% and the two independent predictors of new ischemic lesions were procedural time and IVT treatment. Although I consider this study very interesting, I have some concerns that should be addressed by the authors.
Major concerns
- Lines 57-58: Please, add a table comparing baseline and clinical characteristics between the 78 patients included in the study and the other 61 who were excluded.
- Line 59: Although this study was retrospective and observational, I think that it should be approved by the local Ethics Committee. Please, report the IRB number in the text.
- Lines 64-65: past medical history should include information on previous cerebrovascular events.
- Line 67: Information on use of anticoagulants at admission is important for the aim of this study and it should be reported.
- Lines 70-72: This study was designed for having only one outcome, i.e. the occurrence of procedural thromboembolism. Although the authors reported other outcomes in the text (e.g. mRS, TICI, etc), they treated these variables as independent ones. Thus, the authors should redo the sentence reporting that the occurrence of procedural thromboembolism was the only outcome of this study.
- Lines 104-106: Before using the ROC curves for recognizing the cut-off values of the procedural time for predicting procedural thromboembolism, the authors should use the AUC-ROC curves for assessing whether procedural time and IVT treatment are useful in estimating the occurrence of procedural thromboembolism. Please, report the AUC-ROC value and the 95% CI for each variable, procedural time and IVT, and then for a model that includes both of them.
- Lines 154-158 and Table 2: Please, report in the text and in the table the confounding variables included in the multivariate model.
- Discussion section: The association between procedural thromboembolism and post-procedural hemorrhagic transformation should be discussed.
- Discussion section: The authors should focused their discussion, also, on the lacking association between procedural thromboembolism and poor outcome. It seems that these new ischemic lesions occurred, but they were not able to impair functional status. I consider this acquisition very interesting, thus I suggest the authors to report in the text the anatomical position of each new ischemic lesion after MT.
Minor concerns
- Lines 34-36, sentence “Because of recent…for the treatment of cerebrovascular disease”: Please, add “in patients with large vessel occlusion” at the end of the above reported sentence.
- Line 86: Please, add the appropriate references.
Line 149: Add the appropriate
Reviewer 3 Report
Dear Authors-
Nam et al., is a very interesting study to evaluate the factors of post-MT thromboembolism, only problem is a very small sample size to create a significant conclusion.
Though the concept is very important and crucial to evaluate, some of the risk factors and other factors could come non-significant due to sample size and under-estimate/falsely evaluate the burden. It is highly recommended that authors keep collecting data using the same protocol and present the findings irrespective to current publication status.
Below is my other comments-
Abstract: Mentioned mean time from MT to new-onset AIS
Methods: if a good clinical outcome was defined as a 3-month mRS scores of ≤2 then you may have data of patients up to 3 months, then do you have data of post-MT thromboembolism from days 2 to 3 months? Please add in the methods section why 2 days were considered as a cut-off for DWI and why not longer as 2 days may underestimate the real burden?
If there any criteria you have defined to decide to differentiate new-onset AIS due to thromboembolism or other embolic strokes?
In order to make sample size calculation easy, if possible please add a figure of flow chart how you have included and excluded the patients.
Results: Due to limited sample size of 16 vs 62, many of the risk factors are coming non-significant though might be significant, so it would be good to know the results from bigger sample size if you feel appropriate to continue the study.
Discussion: line 246-248: 4th limitation, you have mentioned late TE are not considered as you have evaluated acute only so please provide evidence why the acute is defined by <=2 days.
Please also add that sample is very small and may underestimate the risk factors and real numbers of stroke in limitation, though you mentioned the need for large study (which is the advance direction).
Thank you
Round 2
Reviewer 2 Report
The present version of the manuscript is clearly improved.
Author Response
Thank you for your consideration in reviewing this manuscript.
Reviewer 3 Report
Dear Authors-
I appreciate the changes you had made.
In Table 2 you used the asterisk in legends but forgot to tag them in the table next to variables.
* Variables for logistic regression model.
Please fix it. Otherwise it’s good to go.
Thank you
Author Response
Thank you for the comment.
We used the asterisk in the table next to the P-value.
We placed the asterisk in the table next to variables, according to your suggestion.
Thank you for your consideration in reviewing this manuscript.